# Physics-Guided Real-Time Full-Field Vibration Response Estimation from Sparse Measurements Using Compressive Sensing

**DOI:** 10.3390/s23010384

**Published:** 2022-12-29

**Authors:** Debasish Jana, Satish Nagarajaiah

**Affiliations:** 1Samueli Civil and Environmental Engineering, University of California Los Angeles, Los Angeles, CA 90095, USA; 2Civil and Environmental Engineering, Rice University, Houston, TX 77005, USA; 3Mechnanical Engineering, Rice University, Houston, TX 77005, USA

**Keywords:** full-field sensing, Compressive Sensing, sparse modelling, physics-guided, full-state estimation, structural health monitoring

## Abstract

In civil, mechanical, and aerospace structures, full-field measurement has become necessary to estimate the precise location of precise damage and controlling purposes. Conventional full-field sensing requires dense installation of contact-based sensors, which is uneconomical and mostly impractical in a real-life scenario. Recent developments in computer vision-based measurement instruments have the ability to measure full-field responses, but implementation for long-term sensing could be impractical and sometimes uneconomical. To circumvent this issue, in this paper, we propose a technique to accurately estimate the full-field responses of the structural system from a few contact/non-contact sensors randomly placed on the system. We adopt the Compressive Sensing technique in the spatial domain to estimate the full-field spatial vibration profile from the few actual sensors placed on the structure for a particular time instant, and executing this procedure repeatedly for all the temporal instances will result in real-time estimation of full-field response. The basis function in the Compressive Sensing framework is obtained from the closed-form solution of the generalized partial differential equation of the system; hence, partial knowledge of the system/model dynamics is needed, which makes this framework physics-guided. The accuracy of reconstruction in the proposed full-field sensing method demonstrates significant potential in the domain of health monitoring and control of civil, mechanical, and aerospace engineering systems.

## 1. Introduction

### 1.1. Motivation of the Study

Due to intense dynamic loads, large amplitude in displacement, velocity, and/or acceleration in the structural systems may induce unanticipated damage to the structures, affecting structural performance. Estimating potentially vulnerable location estimation in the structure requires full-field sensing. Theoretically, nonlinear state estimators [1,2,3] can determine full-field responses from limited sensors. However, these techniques require knowledge of system properties as well as forcing function and the number of limited sensors required for estimating full-state is large. One of the most basic solutions is to install contact-based vibration sensors in a dense manner throughout the structure, but it would be economically expensive and impractical to distribute sensors at all locations. Continuous and distributed sensing is possible with the ‘strain sensing smart skin’ that can be coated onto the surface of the structure to acquire strain data of high spatial resolution for estimating the dynamic strain directly [4,5,6,7,8,9,10]. Recently developed non-contact vision-based techniques like Digital Image Correlation (DIC) [11,12,13,14,15] are very suitable for full-field sensing. In DIC, the target surface, which needs to be monitored, is painted with a random speckle pattern. In recent times, completely non-contact techniques are available for full-field motion estimation, which does not need random speckles on the target surface. Optical flow-based techniques [16,17] track the brightness pattern change in consecutive image frames to estimate the motion, phase-based motion estimation algorithms [18] estimate the oscillatory motion from the local amplitude and phases of the image frames, and the edge tracking [19,20] tracks one particular edge of the structure throughout all the time sample with a sub-pixel level of accuracy. Very recently developed neuro-inspired vision sensors, i.e., event-based cameras  [21] can be used to measure full-field responses. Such cameras are more robust to motion blur, have a higher dynamic range, and have lower latency than a conventional frame-based camera. However, the precision of camera-based techniques depends on the camera’s location and the structure’s size—which sometimes make the camera-based techniques inconvenient when the system is in operating condition. Deep learning [22] can also be deployed for dense response estimation, but it requires extensive training data—which might not always be available.

### 1.2. Contributions

This paper presents a physics-guided approach to determine the full-field vibration response of a structural system from a limited number of randomly placed vibration sensors in a structure. This methodology can be used for contact-based sensors like Linear Variable Differential Transformers (LVDTs) or accelerometers as well as non-contact-based sensors like Laser Doppler Vibrometer (LDV) or computer vision-based point trackers. The time histories of these limited numbers of sensors act as the input of the proposed full-field motion estimation technique. Compressive Sensing (CS) possesses an excellent capability of reconstructing a whole signal from random sampling points. We adopt such potential of the CS method in the spatial domain to generate full-field motion from a sparsely placed limited number of sensors. The spatial basis functions needed for the CS technique are obtained from the closed-form solution of the generalized partial differential equation of the system. It can also be obtained as in a recent work [23] by the authors, wherein Dictionary Learning is utilized to obtain the basis functions—but this requires training data, i.e., dense sensor time history, which may not be unavailable and sometimes infeasible. Therefore, in this paper, we propose a framework to estimate the dense time history from a few randomly placed sensors when the generalized partial differential equation of the system is available. To the best of the authors’ knowledge, no studies are available where full-field vibration response is estimated from limited measurement given the generalized partial differential equation. This framework is physics-guided—as some information of the system’s inherent physics, i.e., the generalized partial differential equation, is required. It is noteworthy that the coefficients of the underlying differential equation are not required; only the overall form/structure of the equation is enough to run the framework. This technique can be applied to the member level as well as the system level of any size of the structure. This method allows the user to execute all the tasks using a limited number of sensors which generally require full-field response, such as (a) full-state estimation, (b) precise localization of potential damage and (c) control force design from full-state feedback.

It is noteworthy that the difference between the recent work by the authors [23] and this paper is that theframework presented in [23] is entirely data-driven, where no information on the system is available; hence, the basis functions are trained using Dictionary Learning from full-field training data. On the other hand, the framework presented in this paper is physics-guided; the information about the generalized partial differential equation of the system is available, and it is also available regarding those scenarios where acquiring the full-field data for training is impossible. As the final objectives of both of these papers ([23] and this article) are similar, we show the results on the same structural systems.

### 1.3. Brief Literature Review on Compressive Sensing

The proposed approach in this paper is motivated by the concept of Compressive Sensing (CS) [24,25,26]. Compressive Sensing techniques can reconstruct a signal from the samples measured below the Nyquist rate, which is essential in data storage and transmission. It uses the sparsity property in the signal to reconstruct the original signal using a few samples. The only challenge of CS is to determine the perfect basis functions for the signals. For example, Fourier and wavelet basis functions are used in standard signal and image compression and transmission as they have sparsity properties in those domains. Similarly, in seismic imaging [27], CS is used in Seismic Image compression as the curvelet domain can be used as the set of basis functions. Most of the application of Compressive Sensing in the domain of structural engineering is related to working with time series data only. In the wireless network, CS has been beneficial for the data packet loss recovery in civil structural health monitoring [28,29]. Using a few measurements in the wireless sensors in the time domain, one technique is proposed to localize structural damage where the matched filter is amalgamated with the CS [30]. In fast-moving wireless sensing, the data-packet loss is common; hence, CS is used to recover the data used for structural health monitoring [31]. As  data compression with CS consumes less power, the long-term bridge monitoring is shown to be energy efficient by 10–60% by using a 20% sample of the original signal [32]. Along with the time series signals, CS is used for efficient and robust data transmission of structural images and videos taken by Unmanned Aerial Vehicles (UAVs) which can be used for machine vision-based SHM applications [33]. Ganesan et al. [34] demonstrate that the CS methodology can also be applied to the spatial domain by reconstructing the operational deflection shape (ODS) of beam vibration from randomly placed sensors. For structural health monitoring purposes, one variant of CS, i.e., Bayesian CS, is used for data loss recovery for approximately sparse signals [35]. Various applications and a detailed review of CS are presented in Rani et al. [36]. In this paper, we implemented the concept of Compressive Sensing in the spatial domain to obtain full-field motion from a few randomly spaced sensors.

The paper is structured as follows. First, the novel framework and the brief concept of Compressive Sensing are shown in Section 2 and Section 3, respectively. Subsequently, numerical simulations on the beam and plate are presented in Section 4. Next, this method is experimentally validated with a video camera on a laboratory-scaled cantilever beam in Section 5. Afterwards, we present the application of this method in determining a potential damage location in Section 6. Finally, some of the important results and conclusions are discussed in Section 7.

## 2. Proposed Formulation Utilizing the Physics-Guided Knowledge

According to the Compressive Sensing theory, a set of basis functions is required to reconstruct a signal from a few measurements. The framework calculates the basis function from the physics-based information of the system. The proposed methodology, in brief, is shown in Figure 1. First, it has to be ensured that a generalized partial differential equation of the mechanical/civil/aerospace system of interest is available. If one is available, then the spatial basis functions are obtained from the closed-form solution of the governing generalized partial differential equation of the system. If the model knowledge is entirely unknown, the basis function can be created from the training data obtained from laboratory testing using Dictionary Learning Algorithm [37,38], which is one of the recent works by the authors [23]. This basis matrix (set of all basis functions) represents the model of the system. This is derived from the generalized partial differential equation of the system—hence, physics-guided. It is noteworthy that only the equations as specified in Figure 1 are enough for this framework to understand the type of systems such as string, beam, plate, or some other structural system. The coefficients of the equations, i.e., $c_{i}$’s, are not required at all—limited knowledge of physics is adequate for this framework. These spatial basis functions contain the system’s intrinsic model knowledge or physics. Second, now for the system of interest, if dense sensing is very difficult to achieve when the structure is in operating condition, then only a few sensors are installed in the structure. In this Figure 1, a sample bridge deck was considered as a system of interest, where four vibration measurement sensors are randomly placed, which records the motion from four locations as shown in the figure. In this figure, only four numbers of actual sensors are shown for demonstration purposes; this number is determined from the basis matrix, which we will discuss in detail later. Finally, the Compressive Sensing technique can estimate the full-field responses using the spatial basis functions and the response history of a small numbers of sensors (in this example, four sensors).

## 3. Full Signal Reconstruction from Few Measurements Using the Concept of Compressive Sensing

The concept of Compressive Sensing [24,25,26] is concisely represented in this section. A signal $y\in R^{m}$ is considered to be sparse, if
(1)$$y=\mathrm{Dx}=\sum_{j=1}^{n}x_{j}d_{j}=\sum_{j\in S}x_{j}d_{j}$$
where $D\in R^{m\times n}$ is the orthonormal basis matrix, $d_{j}$ is the *j*th column of D. Generally, the basis matrix is considered overcomplete, i.e., m<n. Most of the coefficients of $x_{j}$ are zero in Equation (Equation 1), signal sparsity $S=\{j|x_{j}\neq 0\}$, the level of sparsity $s=\left|S\right|={\left||x|\right|}_{0}$; hence, $x\in R^{n}$ is a sparse vector. For completely unknown signals, Dictionary Learning uses the training signals to construct the basis matrices [37,38], but it needs training signals, which sometimes could be unavailable as it needs the full length of $y\in R^{m}$ and optimizing the basis function using Dictionary Learning is indeed computationally expensive. On the other hand, if the intrinsic physics of the system is known, which generates the signals, over-complete basis functions (like Fourier or Wavelet) can be generated.

### 3.1. Solving the Optimization Problem

The Compressive Sensing (CS) algorithm can estimate $y\in R^{m}$ from the noisy measured vector $z\in R^{p}$ with p<<m.
(2)z=Θy+e=ΘDx+e=Φx+e,where,Φ=ΘD
where $\Theta \in R^{p\times m}$ is the measurement matrix. e is the error/noise which is bounded by ${\left||e|\right|}_{2}\leq \epsilon$. Hence, the basis coefficients can be estimated by solving the following convex optimization problem:(3)$$\hat{x}=\underset{||\Phi x-{z||}_{2}\leq \epsilon }{arg min}{\left||x|\right|}_{1}$$
where $\left|\right|\cdot {\left|\right|}_{2}$ denotes the $\ell_{2}$ norm. Equation (Equation 3) can be presented as LASSO [39] framework for optimization as:(4)$$\mathrm{minimize}\;||\Phi x-{z||}_{2}+{\lambda ||x||}_{1}$$
where λ is the regularization parameter. The sparse solution x from Equation (Equation 4) is obtained using the interior point method [40,41], and then the the full signal y can be obtained from Equation (Equation 1).

### 3.2. Minimum Number of Samples Required for Accurate Signal Reconstruction

Amini et al. [42] suggest the minimum number of the sampling points for accurate signal reconstruction is dependent on the basis matrix and can be estimated as follows:Perform Singular Value Decomposition (SVD) of D as D=UΣV, where $D\in R^{m\times n}$, $U\in R^{m\times m}$, $V\in R^{n\times n}$, and $\Sigma \in R^{m\times n}$ with m<n. Diagonal values of Σ represent the singular values, and $\sigma_{i}$ represents the *i*th singular value.Normalized Power Index (NPI) or the expressivity index is established as ${\mathrm{NPI}}_{p}=\frac{\sum_{i=1}^{p}\sigma_{i}^{2}}{\sum_{i=1}^{m}\sigma_{i}^{2}}$. The minimum sensor number for accurate signal reconstruction is the minimum integer value of *p* for which NPI→1.

## 4. Numerical Studies

In this paper, we numerically demonstrated the results of a 1D system (beam) and a 2D system (plate). For each of the cases, the basis matrix $D\in R^{m\times n}$ is obtained from the inherent physics, specifically from the closed-form solution of the governing generalized partial differential equation of the continuous system. This methodology can easily be extended to other continuous systems like strings, membranes, etc.

### 4.1. Numerical Studies on a Simply Supported Beam

#### 4.1.1. System Properties

A simply supported continuous steel beam [23] of length, height, and width of 50 m, 1 m, and 0.5 m, respectively, is considered as shown in Figure 2. The system is assumed to have 1% Rayleigh Damping.

#### 4.1.2. Working Procedure

In this numerical example, the virtual sensing points are considered to be spaced at 0.05 m, making the total number of such points 999. The number of virtual sensing points or the distance between them can be modified based on user specifications. The number of randomly placed actual vibration sensors is 10 (indicated as red circles in Figure 2), which is very few compared to 999 (denoted as blue dots in Figure 2).

A random forcing function of low-frequency content (0–20 Hz) is applied 10 m from the left end of the beam to induce the vibration in the simply supported continuous beam. This ambient random forcing function’s mean and standard deviations are zero and 100 N, respectively. In this paper, the Finite Element Method (FEM) was used to numerically estimate the vibration responses of the beam due to this random ambient force for 12 s with a sampling frequency of 1000 Hz. Note that the number of virtual sensing points, the type and amplitude of forcing functions, and the sampling frequency are similar to the examples presented in a previous work by the authors [23].

#### 4.1.3. Basis Function Creation from Physics-Based Knowledge

The proposed framework requires the estimation of basis matrix D from the closed-form solution of the inherent differential equation of the continuous system.

Transverse vibrations of beams for broadband random excitation:

The equation of motion of a Euler–Bernoulli beam excited by a distributed transverse force can be expressed as [43]
(5)$$\frac{\partial^{2}}{\partial x^{2}}\left[EI\left(x\right)\frac{\partial^{2}w\left(x,t\right)}{\partial x^{2}}\right]+\rho A\left(x\right)\frac{\partial^{2}w\left(x,t\right)}{\partial t^{2}}=f\left(x,t\right)$$
where w(x,t) is the transverse response of the beam, f(x,t) represents the forcing function, *E* is Young’s Modulus, ρ is the density, I(x) and A(x) are the moment of inertia and cross-sectional area at distance *x* from one end, respectively. The transverse response can be assumed to be a linear combination of the normal modes of the beam as
(6)$$w\left(x,t\right)=\sum_{i=1}^{\infty }W_{i}\left(x\right)\eta_{i}\left(t\right)$$
where the *i*th mode is represented by the mode shape $W_{i}\left(x\right)$ in generalized coordinates. The solution of Equation (Equation 5) is expressed as
(7)$$w\left(x,t\right)=\sum_{i=1}^{\infty }\left[A_{i}\cos\omega_{i}t+B_{i}\sin\omega_{i}t+\frac{1}{\omega_{i}}\int_{0}^{t}Q_{i}\left(\tau \right)\sin\omega_{i}\left(t-\tau \right)d\tau \right]W_{i}\left(x\right)$$
where $Q_{i}\left(t\right)$ is the generalized force corresponding to the *i*th mode as:(8)$$Q_{i}\left(t\right)=\int_{0}^{l}W_{i}\left(x\right)f\left(x,t\right)dx$$

The first two terms of Equation (Equation 7) are due to the free vibration, and the constants $A_{i}$ and $B_{i}$ are evaluated from the initial conditions; the third term denotes the forced vibration.

For beams with a uniform cross-section, the mode shapes $W_{i}\left(x\right)$ are functions of natural frequencies $\omega_{i}$. A general expression of the mode shape is expressed as:(9)$$W_{i}\left(x\right)=C_{i}\cos\beta_{i}x+D_{i}\sin\beta_{i}x+E_{i}\cosh\beta_{i}x+F_{i}\sinh\beta_{i}x$$
where the spatial parameter β is related to ω as $\omega =\beta^{2}\sqrt{\frac{EI}{\rho A}}$. The constants $C_{i}$, $D_{i}$, $E_{i}$, and $F_{i}$ for the *i*th mode depend on the boundary conditions. For beams with varying cross-sectional area, such general and closed-form expression of mode shape can be found in the literature [44,45,46,47].

Spatial Sparsity:

Operational Deflection Shape (ODS) can be used for damage localization of a beam. A dense deflection shape can be reconstructed using the discrete sensor placed in a randomized location on the beam, eliminating the optimal sensor location determination. ODS of beams are also sparse but in the spatial frequency domain. The mode shape functions depend on the spatial parameter β, which uniquely relates to the natural frequencies ω. Hence, Equation (Equation 7) can be expressed as follows    
(10)$$\begin{array}{cc} w\left(x,t\right)=\sum_{i=1}^{\infty }W_{i}\left(x\right)\eta_{i}\left(t\right) & =\sum_{i=1}^{\infty }\left(C_{i}\cos\beta_{i}x+D_{i}\sin\beta_{i}x+E_{i}\cosh\beta_{i}x+F_{i}\sinh\beta_{i}x\right)\eta_{i}\left(t\right)\\ \; & =\sum_{i=1}^{\infty }\left({\tilde{C}}_{i}\cos\beta_{i}x+{\tilde{D}}_{i}\sin\beta_{i}x+{\tilde{E}}_{i}\cosh\beta_{i}x+{\tilde{F}}_{i}\sinh\beta_{i}x\right) \end{array}$$
where ${\tilde{C}}_{i}=C_{i}\eta_{i}\left(t\right),{\tilde{D}}_{i}=D_{i}\eta_{i}\left(t\right),{\tilde{E}}_{i}=E_{i}\eta_{i}\left(t\right),{\tilde{F}}_{i}=F_{i}\eta_{i}\left(t\right)$. The range of spatial frequency parameter β can be determined from the range of natural frequency ω as they are directly related. Now, the basis matrix $D\in R^{m\times n}$ is created from the range/length of the beam and the range of spatial frequency. Here, the location on the beam $x_{k}=\frac{L\cdot (k-1)}{m-1},k=1,2,\cdots m$ and the spatial frequency $\beta_{k}=\beta_{low}+\frac{\left(\beta_{high}-\beta_{low}\right)}{\left(n-1\right)}\cdot \left(k-1\right),k=1,2,\cdots ,n$ with the spatial frequency range $\beta_{range}=\left[\beta_{low},\beta_{high}\right]$. Now, from the ranges mentioned in the previous sentence and Equation (Equation 10), the basis matrix $D\in R^{m\times n}$ is obtained using the cosine, sine, hyperbolic cosine and hyperbolic sine basis. D is expressed as the concatenation of $\Delta_{1}$ and $\Delta_{2}$ as
(11)$$D=[\Delta_{1},\Delta_{2}]$$
where
$$\Delta_{1}=\left[\begin{array}{cccccc} \cos(\beta_{1}x_{1}) & \cdots & \cos(\beta_{n}x_{1}) & \sin(\beta_{1}x_{1}) & \cdots & \sin(\beta_{n}x_{1})\\ \cos(\beta_{1}x_{2}) & \cdots & \cos(\beta_{n}x_{2}) & \sin(\beta_{1}x_{2}) & \cdots & \sin(\beta_{n}x_{2})\\ \vdots & \vdots & \vdots & \vdots & \vdots & \vdots \\ \cos(\beta_{1}x_{m}) & \cdots & \cos(\beta_{n}x_{m}) & \sin(\beta_{1}x_{m}) & \cdots & \sin(\beta_{n}x_{m}) \end{array}\right]$$
$$\Delta_{2}=\left[\begin{array}{cccccc} \cosh(\beta_{1}x_{1}) & \cdots & \cosh(\beta_{n}x_{1}) & \sinh(\beta_{1}x_{1}) & \cdots & \sinh(\beta_{n}x_{1})\\ \cosh(\beta_{1}x_{2}) & \cdots & \cosh(\beta_{n}x_{2}) & \sinh(\beta_{1}x_{2}) & \cdots & \sinh(\beta_{n}x_{2})\\ \vdots & \vdots & \vdots & \vdots & \vdots & \vdots \\ \cosh(\beta_{1}x_{m}) & \cdots & \cosh(\beta_{n}x_{m}) & \sinh(\beta_{1}x_{m}) & \cdots & \sinh(\beta_{n}x_{m}) \end{array}\right]$$

Now, from the location of *p* number of randomly placed sensors, the measurement matrix $\Theta \in R^{p\times m}$ is obtained, and in a similar fashion, Φ in Equation (Equation 2) will be randomly chosen rows of basis matrix **D**. For *p* number of measurement, z=Φx and the sparse solution $x={\left[C_{1}^{*},\cdots ,C_{n}^{*},D_{1}^{*},\cdots ,D_{n}^{*},E_{1}^{*},\cdots ,E_{n}^{*},F_{1}^{*},\cdots ,F_{n}^{*}\right]}^{T}$ can be obtained from $\ell_{1}$ minimization.

For a simply supported beam with length *L*, the deflection equation in Equation (Equation 10) is expressed as
(12)$$w\left(x,t\right)=\sum_{i=1}^{\infty }W_{i}\left(x\right)\eta_{i}\left(t\right)=\sum_{i=1}^{\infty }D_{i}\sin\left(\beta_{i}x\right)\eta_{i}\left(t\right)=\sum_{i=1}^{\infty }D_{i}\sin\frac{i\pi x}{L}\eta_{i}\left(t\right)=\sum_{i=1}^{\infty }{\tilde{D}}_{i}\sin\frac{i\pi x}{L}$$

Hence, the basis matrix in Equation (Equation 11) will be reduced to
(13)$$D=\left[\begin{array}{ccc} \sin(\beta_{1}x_{1}) & \cdots & \sin(\beta_{n}x_{1})\\ \sin(\beta_{1}x_{2}) & \cdots & \sin(\beta_{n}x_{2})\\ \vdots & \vdots & \vdots \\ \sin(\beta_{1}x_{m}) & \cdots & \sin(\beta_{n}x_{m}) \end{array}\right]$$

With *p* random measurement along the length of the beam for a definite time instant can be written as [34]: $z_{j}=\sum_{q=1}^{n}D_{q}^{*}\sin\left(\beta_{q}x_{j}\right);j=1,2,\cdots ,p$ and concisely can be written in a matrix form z=ΘDx=Φx. Here, $x={\left[D_{1}^{*},D_{2}^{*},\cdots ,D_{n}^{*}\right]}^{T}$ and the expected sparse solution should have non-zero $D_{q}^{*}$ if $D_{q}^{*}\approx {\tilde{D}}_{i}$. One important thing to note is that the sparse coefficients $D_{q}^{*}$ are different than the basis matrix $D\in R^{m\times n}$, and the sparse solution x is different than the spatial locations $x_{i}$.

In this numerical example, the number of spatially dense virtual sensors is considered to be 999, and they are equidistantly placed over the length of the beam. Considering the two endpoints as boundaries, the number of rows *m* in the basis matrix is 1001. The range of spatial frequency can be obtained from the natural frequency, and the range of ω can easily be obtained from the Fourier transform of response. Here, the range of spatial frequency is chosen as $\left[\beta_{low},\beta_{high}\right]=\left[0.001,0.3\right]$. The spatial frequency range is discretized into n=300 divisions, which makes the size of the basis matrix $D\in R^{1001\times 300}$.

#### 4.1.4. Minimum Number of Actual Sensors Needed for Accurate Reconstruction

According to the specifications presented in Section 3.2, the optimal number of sensors required for accurate reconstruction is dependent on the SVD of physics-guided basis matrix D. The SVs and corresponding NPIs are shown in Figure 3, and it is observed that the minimum sensor number for accurate reconstruction should be 7. We will discuss both the result from 7 and 10 sensors for this case.

#### 4.1.5. Response Estimation of Densely Located Virtual Sensing Points Using Compressive Sensing

From the physics-guided basis matrix, we performed Compressive Sensing for the whole time series to obtain the time history of all the virtual sensing points. The resulting full-field time history using the proposed framework is compared with the exact time history obtained from the finite element formulation. A relative error metric $\epsilon_{i}$ [23] is considered to address the effectiveness of the proposed method:(14)$$\epsilon_{i}=\frac{\left|\right|R_{\mathrm{Exact},i}-R_{\mathrm{Estimated},i}{\left|\right|}_{2}^{2}}{\frac{1}{m}\sum_{i=1}^{m}\left|\right|R_{\mathrm{Exact},i}{\left|\right|}_{2}^{2}}\times 100;\;i=\mathrm{sensor}\;\mathrm{index}$$
where $\left|\right|\cdot {\left|\right|}_{2}$ denotes the 2-norm and *m* is the number of virtual sensing points. $R_{\mathrm{Estimated},i}$ and $R_{\mathrm{Exact},i}$ denote the estimated response and the exact response of the *i*th virtual sensing point. Both $R_{\mathrm{Exact},i}$ and $R_{\mathrm{Estimated},i}$ are of dimension $n_{t}\times 1$, $n_{t}$ is the number of time samples. The overall average error E [23] is represented as:(15)$$E=\frac{1}{n_{s}}\sum_{i=1}^{n_{s}}\epsilon_{i}$$
where E is the mean error of all the relative errors $\epsilon_{i}$ of independent virtual responses, and hence invariant to the number of virtual sensors.

In this study, FEM is used to calculate the virtual dense sensing point responses for 12 s. Here, the first 2 s of the data are used as validation data to find the optimal hyperparameter λ in Equation (Equation 4), which governs the sparsity of the problem. In an experimental scenario, such λ can be found from the numerical simulation of the system model. The last 10 s of data are used for the testing. The sampling frequency is 1000 Hz; hence, the time samples for hyperparameter optimization (validation) and testing are 2000 and 10,000, respectively. The accuracy of the framework is reported only on the test data. λ is a very robust hyperparameter, and it does not require fine-tuning as the average error is not very sensitive to a slight change of λ. The average validation error E values for different hyperparameter λ is shown in Figure 4. The optimal λ is chosen as 0.001 where the average error is minimum in Figure 4.

$\epsilon_{i}$ is estimated for each dense virtual point for the test data (2–12 s), and the error profile for the full beam length is shown in Figure 5a. The average test error E for all the virtual sensors $7.2\times 10^{-4}\%$ indicates the accuracy of the proposed technique. The largest $\epsilon_{i}$ is 0.0025% located at 4.3 m from the left end and denoted as Location 1 in Figure 5a,b. The maximum error over the beam is so small, which indicates the proposed framework can successfully obtain the full-field time history from a handful of actual sensors. The reconstructed time history of Location 1 is plotted in Figure 5c, along with the actual time history, showing the effectiveness of the proposed method. Notably, this approach’s accuracy is invariant to the location of the forcing function.

#### 4.1.6. Practical Recommendations on the Actual Sensor Placement

In Section 4.1.5, the actual sensors were considered to be placed in a completely random fashion. However, it might be possible that in the random placement of sensors, two or more sensors are located within a minimal distance, and the user does not want such clustering of sensors, e.g., in Figure 5, the leftmost two sensors are very close to each other. Hence, to avoid such scenarios, sensors can be placed using Latin Hypercube Sampling (LHS) or stratified random sampling instead of pure random sampling. LHS is also a variant of random sampling, which satisfies the RIP property. In this case, we assume that *p* number of actual sensors has to be installed in a stratified random fashion on a beam of length *L* [23].

#### 4.1.7. Full-Field Response Estimation from the Minimum Number of Actual Sensors

Responses of all 999 virtual sensing points are calculated from optimally obtained seven sensors (presented in Section 4.1.4) placed in a stratified random manner using the proposed framework. With the optimally estimated seven sensors (Section 4.1.4) placed in a stratified random manner, responses are estimated for all 999 virtual sensing points using the proposed framework. The error profile is shown in Figure 6a, and the average test error is 0.0011%. The maximum error is 0.0045% over the full beam length and is at 35.1 m from the left end. However, the amplitude of the maximum error is negligible, and the estimated time history of this point perfectly overlaps with the true/exact time history as shown in Figure 6c, demonstrating the capability of the framework.

#### 4.1.8. Effect of Measurement Noise on the Full-Field Response Estimation

If the actual sensors that are installed in the structure contain the measurement noise, then the average full-field response estimation will be affected. Hence, the proper noise filtering technique such as bandpass filters can be utilized to at least get rid of the high frequency noises, prior to using those responses for full-field response estimation. However, in this section, the effect of measurement noise on the framework is shown if no noise filtering technique were used. In these numerical simulations, the added measurement noise sequences are obtained as r% root mean square (RMS) zero mean Gaussian white noise, with the percentage computed with respect to the RMS of the corresponding true response of the attached sensors as shown in Figure 6. The estimation accuracy is calculated for four different noise levels, for *r* as 1, 2, 5, and 8. Beyond these noise levels, the sensors could be considered as faulty and might need replacement. The average estimation errors for these noise levels are shown in Figure 7 for the test data of 10 s. It is obvious that with the increasing measurement noise level, the full-field response estimation error would increase. However, the small error value justifies the robustness of the proposed framework for the measurement noise. As the measurement noise sequences are random and cannot be represented by the spatial basis functions, the effects of such noises in Compressive Sensing-based frameworks are nominal.

### 4.2. Numerical Studies on Simply Supported Rectangular Plate

#### 4.2.1. System Properties

A steel plate [23] of length (L) 3 m, width (B) 2 m, and thickness (T) 0.005 m is considered. The system is assumed to have 1% Rayleigh Damping. To induce vibration in the plate, a random dynamic force of frequency content 0–20 Hz is utilized to actuate a point that is 0.8 m from the bottom end and 1 m from the left end. The random force’s mean and standard deviation is zero and 100 N, respectively. It is worth noting that the system properties and the characteristics of the random force are the same as the example presented in a previous work by the authors [23]. In this numerical study, the response due to this dynamic force is calculated for 12 s, and the sampling frequency is 1000 Hz. Similar to the numerical simulation of the simply supported beam, the first 2 s (2000 time samples) of data are used for the hyper-parameter tuning (validation), and the remaining 10 s of data (10,000 time samples) are used for testing.

#### 4.2.2. Basis Function Creation from Physics-Guided knowledge

The equation of motion of the thin plates or classical/Kirchhoff plates, which is based on similar assumptions as a thin beam or Euler–Bernoulli beam excited by a distributed transverse force, can be expressed as [43]
(16)$$D\left(\frac{\partial^{4}w\left(x,y,t\right)}{\partial x^{4}}+2\frac{\partial^{4}w\left(x,y,t\right)}{\partial x^{2}\partial y^{2}}+\frac{\partial^{4}w\left(x,y,t\right)}{\partial y^{4}}\right)+\rho h\frac{\partial^{2}w\left(x,y,t\right)}{\partial t^{2}}=f\left(x,y,t\right)$$
where w(x,y,t) is the transverse response of the plate, f(x,y,t) represents the distributed transverse forcing function acting on the plate per unit area, *h* is the plate thickness, ρ is the density, *D* represents the flexural rigidity of the plate and is expressed as $D=\frac{Eh^{3}}{12(1-\nu^{2})}$, where *E* is Young’s Modulus and ν is the Poisson ratio of the plate.

The transverse displacement response of the plate can be expressed as the linear combination of the normal modes of the plate as
(17)$$w\left(x,y,t\right)=\sum_{i=1}^{\infty }\sum_{j=1}^{\infty }W_{ij}\left(x,y\right)\eta_{ij}\left(t\right)$$

For a general plate, the normal modes are expressed as
(18)$$\begin{array}{cc} W(x,y)= & C_{1}\sin\gamma_{1}x\sin\gamma_{2}y+C_{2}\sin\gamma_{1}x\cos\gamma_{2}y+C_{3}\cos\gamma_{1}x\sin\gamma_{2}y\\ \; & +C_{4}\cos\gamma_{1}x\cos\gamma_{2}y+C_{5}\sinh\gamma_{3}x\sinh\gamma_{4}y+C_{6}\sinh\gamma_{3}x\cosh\gamma_{4}y\\ \; & +C_{7}\cosh\gamma_{3}x\sinh\gamma_{4}y+C_{8}\cosh\gamma_{3}x\cosh\gamma_{4}y \end{array}$$

For a simply supported rectangular plate, Equation (18) is simplified to
(19)$$w\left(x,y,t\right)=\sum_{i=1}^{\infty }\sum_{j=1}^{\infty }\left(C_{1ij}\sin\frac{i\pi x}{L}\sin\frac{j\pi y}{B}\right)\eta_{ij}\left(t\right)=\sum_{i=1}^{\infty }\sum_{j=1}^{\infty }{\tilde{C}}_{1ij}\sin\frac{i\pi x}{L}\sin\frac{j\pi y}{B}$$

Hence, for this case, the vibration response for a bounded number of points along the 2D domain of the plate for a definite time instant can be expressed as
(20)$$y_{ij}=\sum_{r=1}^{R}\sum_{q=1}^{Q}C_{rq}^{*}\sin\left(\zeta_{r}x_{i}\right)\sin\left(\xi_{q}y_{j}\right);\;y=Dx;$$
where $x=\Delta_{3}⊗\Delta_{4}$; here, ⊗ denotes the Kronecker tensor product between matrices. The two-dimensional matrix D is of size $\left((I\times J)\times (R\times Q)\right)$ with (I×J) elements in the rows and (R×Q) elements in the columns.
(21)$$\Delta_{3}=\left[\begin{array}{ccc} \sin(\zeta_{1}x_{1}) & \cdots & \sin(\zeta_{R}x_{1})\\ \sin(\zeta_{1}x_{2}) & \cdots & \sin(\zeta_{R}x_{2})\\ \vdots & \vdots & \vdots \\ \sin(\zeta_{1}x_{I}) & \cdots & \sin(\zeta_{R}x_{I}) \end{array}\right];\;\Delta_{4}=\left[\begin{array}{ccc} \sin(\xi_{1}y_{1}) & \cdots & \sin(\xi_{Q}y_{1})\\ \sin(\xi_{1}y_{2}) & \cdots & \sin(\xi_{Q}y_{2})\\ \vdots & \vdots & \vdots \\ \sin(\xi_{1}y_{J}) & \cdots & \sin(\xi_{Q}y_{J}) \end{array}\right]$$
where $\zeta_{i}$ and $\xi_{j}$ are the spatial frequency along the *x* and *y* direction of the plate, respectively.

Now, the basis matrix D is created from the range/length of the plate and the range of spatial frequency in both directions. Here, *x* positions on the plate are denoted as $x_{i}=\frac{L\cdot (i-1)}{I-1},\;i=1,2,\cdots I$, the *y* positions are denoted as $y_{j}=\frac{B\cdot (j-1)}{J-1},\;j=1,2,\cdots J$. The spatial frequency in the *x* direction is represented as $\zeta_{r}=\zeta_{low}+\frac{\left(\zeta_{high}-\zeta_{low}\right)}{\left(R-1\right)}\cdot \left(r-1\right),\;r=1,2,\cdots ,R,$ with the spatial frequency range in *x* expressed as $\zeta_{range}=\left[\zeta_{low},\zeta_{high}\right]$; similarly, spatial frequency in *y* is represented as $\xi_{q}=\xi_{low}+\frac{\left(\xi_{high}-\xi_{low}\right)}{\left(Q-1\right)}\cdot \left(q-1\right),\;q=1,2,\cdots ,Q,$ with the spatial frequency range in *y* expressed as $\xi_{range}=\left[\xi_{low},\xi_{high}\right]$.

*p* random measurements on the plate for a definite time instant can be written concisely in a matrix form z=ΘDx=Φx. Here, $x={\left[C_{1}^{*},C_{2}^{*},\cdots ,C_{\left(R\times Q\right)}^{*}\right]}^{T}$ and the expected sparse solution should have non-zero $C_{rq}^{*}$ if $C_{rq}^{*}\approx {\tilde{C}}_{1ij}$. One important thing to note is that the plate motion amplitude y is different than the y-coordinates in plate $y_{j}$, and the sparse solution x is different than the spatial locations $x_{i}$.

In this numerical example, the number of gridpoints along *x* and *y* are considered as I=31 and J=21, respectively. Moreover, the ranges in spatial frequencies are $\left[\zeta_{low},\zeta_{high}\right]=\left[0.1,7\right]$ and $\left[\xi_{low},\xi_{high}\right]=\left[0.1,10\right]$ with R=70 and Q=100. These values are obtained from the natural frequency, assuming that the response will be limited to the first six modes of vibration. This makes the size of the basis matrix $D\in R^{651\times 7000}$.

#### 4.2.3. Response Estimation of Dense-Grid of Virtual Sensing Points Using Compressive Sensing

SVD is performed on the physics-guided basis matrix D and SVs and corresponding NPIs are shown in Figure 8a,b, respectively. Most of the dominant singular values are the first 36 singular values, even though the NPIs for the 36 sensors are 0.8537, which is not very close to 1; in this numerical study, 36 sensors are chosen for full-field reconstruction, as incorporating more sensor will incur more cost.

We performed Compressive Sensing for the whole time series using the physics-guided basis matrix. For each of the time instances, CS estimates virtual dense sensing point response time history from 36 actual sensors placed in a stratified random manner, which is shown in Figure 9a. Using the first 2 s of response, the tuned hyperparameter λ is found to be 0.1. Now, the relative errors $\epsilon_{i}$ are estimated for all virtual sensing points for the test dataset, shown in Figure 9b—the average test error E is 0.35%. Relatively significant errors are estimated as 5.13% and 4.48% (denoted as Locations 1 and 2 in Figure 9c), and their time history comparison is presented in Figure 10. With 36 actual sensors, full-field sensing of 2D continuous systems are possible, and increasing the number of actual sensors will lead to more accurate estimation.

## 5. Experimental Verification

Experimental validation of the proposed framework using contact-based sensors is a bit difficult as it is practically impossible to mount dense sensors over any structure. Dense sensor data will be needed to validate the reconstructed response from the randomly spaced actual sensors. Apart from being economically inefficient, mounting a dense array of sensors on a structure will modify the whole system’s mass properties, altering the inherent system’s true dynamic response. Hence, we adopted a camera-based sensing technique called edge tracking, which outputs the responses of each pixel present in the edge, making it a full-field response estimation technique. For perfect estimation of the responses, the edge tracking algorithm is expected to have accuracy at the subpixel level.

### 5.1. Edge Detection at Subpixel Level

In this paper, the Subpixel Edge Location Algorithm [48] was utilized to acquire the edge information accurately. This algorithm localizes the edges by extracting the edges’ position, orientation, curvature, and contrast using one digital image acquisition technique, the partial area effect. A brief concept [23] is presented in Figure 11.

### 5.2. Vibration Data Acquisition Using Video Camera

We performed one dynamic experiment on a laboratory-scaled aluminum cantilever beam [23] to demonstrate the capability of the proposed technique. The experimental set-up is shown in Figure 12. This specimen’s length, breadth, and depth are 20.25, 2, and 1/32 inches, respectively. To induce dynamic vibration in the beam, we simulated the ambient wind load with an air blower (Figure 12). Please note that the experimental set-up is the same as in a previous study by the authors [23].

During the whole time of the vibration experiment, we tracked the beam’s whole edge using the subpixel level of accuracy with the partial area effect algorithm [48], and each pixel on the edge of the beam acted as virtual sensors. The final objective is to validate the measured pixel displacement time histories with the reconstructed time history obtained from a handful of measurements. This limited number of measurements represents actual sensors in this experiment which are some random pixels on the edge itself.

We use the camera of the Apple iPhone SE (Figure 12) to capture the video of the cantilever beam vibration. iPhone’s in-built slow-motion mode allows us to capture the video at a frame rate of 240 with a resolution of 720p (frame height × width = 1280×720 pixels). The vibration of the cantilever beam video was recorded for 35 s. The camera was placed at 254 mm (10 inches) from the specimen. The fixed support constrains the observable length of beam vibration by the iPhone as 501.65 mm (19.75 inches).

### 5.3. Processing the Recorded Video

In this section, we discuss the procedure to calculate the response time history of each edge pixel from the recorded video. First, the video is cropped so that the region only focuses on the cantilever edge. Each of the frames in the RGB video is converted to grayscale as the colour information is redundant in edge processing. Cropping and grayscale conversion reduce the physical dimensions of the images/video frames, making the overall edge processing computationally efficient. Processed edges from cropped and grayscaled video frames of one instance are shown in Figure 13.

The image pixel units are converted to physical displacement units, and the scaling unit was found to be 36 pixels/inch (≈1.417 pixels/mm) to convert the displacement time histories from pixel to displacement unit. A low-pass filter cutoff of 30 Hz is applied to each of the time histories to eliminate high-frequency noises. The cantilever consists of 711 pixels (711/36 = 19.75 inches or 501.65 mm), which represents 711 dense virtual sensing points on the cantilever. The response time history near the support is erroneous from the camera-based measurement as the vibration amplitude adjacent to the support is tiny, which makes the Signal to Noise Ratio (SNR) very low. Due to this erroneous measurement, comparing the estimated response from the framework with the true response (low SNR) would not make any sense. Therefore, only the top part of the beam, i.e., 15.45 inches or 392.43 mm from the cantilever tip or top 556 pixels (556/36 = 15.45 inches) are compared with the reconstructed time histories; we do not consider the bottom 4.3 inches (19.75 − 15.45 = 4.3) or 109.22 mm. The video lasts 35 s, and there are a total of 8400 frames, as the fps rate is 240. The first 10 s (2400 time samples) of data are used for the hyper-parameter optimization, and the remaining 25 s (6000 time samples) are used for testing.

### 5.4. Basis Function Creation from Physics-Guided Knowledge

The Euler–Bernoulli cantilever beam follows the differential equation expressed in Equation (Equation 5), and the general *i*th mode shape of the cantilever beam is expressed as follows with $G_{i}$ as a constant [43]
(22)$$W_{i}\left(x\right)=G_{i}\left[\left(\cos\beta_{i}x-\cosh\beta_{i}x\right)-\frac{\cos\beta_{i}l+\cosh\beta_{i}l}{\sin\beta_{i}l+\sinh\beta_{i}l}\left(\sin\beta_{i}x-\sinh\beta_{i}x\right)\right]$$

Hence, the overall spatiotemporal deflection profile of the cantilever can be expressed as follows.
(23)$$w\left(x,t\right)=\sum_{i=1}^{\infty }W_{i}\left(x\right)\eta_{i}\left(t\right)=\sum_{i=1}^{\infty }\left({\tilde{C}}_{i}\cos\beta_{i}x+{\tilde{D}}_{i}\sin\beta_{i}x+{\tilde{E}}_{i}\cosh\beta_{i}x+{\tilde{F}}_{i}\sinh\beta_{i}x\right)$$

Hence, for the Compressive Sensing framework, the basis function $D\in R^{m\times n}$ will be the concatenation of $\Delta_{1}$ and $\Delta_{2}$ as $D=[\Delta_{1},\Delta_{2}]$, where the expressions of $\Delta_{1}$ and $\Delta_{2}$ are given in Equation (Equation 11). We worked with the top 556 pixels of the beam; the number of spatial points m=556. Here, the range of spatial frequency is chosen as $\left[\beta_{low},\beta_{high}\right]=\left[0.001,0.35\right]$ and the spatial frequency range is discretized into n=350 divisions, which makes them the size of the basis matrix as $D_{top}\in R^{556\times 1400}$.

### 5.5. Response Estimation of Densely Located Virtual Sensing Points (Image Pixels on the Beam) Using Compressive Sensing

Similar to all the previous examples, the minimum number of actual sensors needed for full-field sensing for accurate reconstruction is estimated by performing the SVD of the physics-guided basis matrix $D_{top}\in R^{556\times 1400}$. The SVs of D and corresponding NPIs are shown in Figure 14. The minimum sensor number is found to be 6.

Compressive Sensing is performed for all the test data using the physics-guided basis matrix and the optimal number of stratified randomly placed sensors as shown in Figure 15a. Using the optimized hyperparameter λ as 0.01 obtained from the validation dataset, the vibration responses of all the virtual sensing points is estimated for the test dataset. The relative error profile is shown in Figure 15b. The average test error E is 0.017%—indicating the accuracy of the proposed method. The three largest errors are 0.096%, 0.059%, and 0.057% denoted as Locations 1, 2, and 3 in Figure 15c. The time histories for these three locations are shown in Figure 16 to depict the comparison with the actual response. It is to be noted that, in this experiment, the external force was imparted by an air blower; hence, there was no control over the amplitude of the force amplitude and the point of actuation. The proposed methodology is invariant to the properties of the external forcing function; therefore, the reconstructed full-field time histories are accurate and comparable with the measured time history. In this article, the performance of the proposed framework is presented and validated on the video of cantilever beam vibration; this can be adapted for cable type structures also [49,50].

## 6. Potential Applications of the Proposed Framework

The proposed framework estimates the full-field response time histories from a handful of sensors in real time for a partially unknown dynamical system. Here, we define the system as ‘partially unknown’ because, even though the form of the generalized partial differential equation of the system is known, the parameters are unknown. Hence, applications of the proposed technique could be (i) estimating the system parameters of the generalized partial differential equation [51,52,53], (ii) potential damage location determination (full-field responses can be decomposed into full-field mode shapes which can be used for damage localization [54]), (iii) full-state estimation of moving targets [55] with parametric variation [56,57], (iv) control force estimation from full-state feedback [58]. These state feedback-based active control techniques are applied in acoustic impedance [59], vehicle suspension [60], and vibration control [61]. Not only in active control, but the state feedback-based control is also utilized in semi-active control [62], and hybrid control [63] also. Conventionally, for system identification and damage localization tasks, the position of sensors (outputs) and the forces (inputs) can be optimally estimated by optimizing the Bayesian loss function [64], modal kinetic energy [65], Fisher Information Matrix (FIM) norms [66,67,68,69] or information entropy [70,71]. This article mainly discusses the implementation of a full-field response estimation strategy for structural elements—the proposed framework can also be extended in fluid-structure interaction problems, e.g., wind pressures measurement [72,73] on building from a few sensors. The full-field response estimation capability will allow for achieving the dense operational deflection shape of the system with fewer sensors—which could be used for structural damage detection and localization [74,75,76]. As this technique works in real time, this technique can easily be utilized for online health monitoring of civil engineering structures and rotating machinery [77,78,79,80].

It is to be noted that this paper formulates the full-field time history estimation from a few sensors as a data sparsity in the spatial domain problem. If the few actual sensors mounted on the system also suffer from data sparsity in the time domain, which is very common in wireless sensors due to packet loss [28,81,82], then also this proposed framework cannot applicable after a pre-processing stage. In this pre-processing step, Compressive Sensing can be adopted to reconstruct the time series of the packet data lost signal, and for the vibration measurements, sinusoidal (sine and cosine) functions can be considered as the basis functions. In an online estimation scenario, window-based Compressive Sensing [29] can be utilized to obtain the reconstructed time series in real time. This proposed technique is physics-guided; the generalized partial differential equation is needed for this approach to work, which is mostly available for simple systems/structures. Hence, for complex and unconventional 3D structural systems, the proposed framework cannot be directly applicable. In those scenarios, one of the approaches could be dividing the complex system into simple substructural systems, where the physics-based equations are available, and then deploying the proposed technique. Another alternative option would be estimating the basis functions for the complex structure as a whole and learning the spatial basis functions from the training data using the Dictionary Learning technique [23].

In this paragraph, we demonstrate that full-field sensing will provide information about possible damage locations, which is impossible to find with a limited number of sensors. The beam discussed in Section 4.1 has a maximum displacement at 33.25 m from the left end, denoted as a red dot in Figure 17 over the full-time period for the given random force. If no actual sensor is present at that location, it would be impossible to figure out such locations.

## 7. Conclusions

Accurate full-field measurement necessitates dense sensing, which often could be uneconomical as well as unfeasible. In this paper, we proposed a framework that requires a handful of contact/non-contact-based sensors to determine the full-field response for a partially unknown dynamical system. The proposed methodology uses the Compressive Sensing technique to estimate the instantaneous spatial motion profile from the randomly placed actual sensors (contact/non-contact based), and iterating this procedure for all time steps, we can obtain a full-field spatiotemporal profile of any vibrating structure. The basis functions of Compressive Sensing are created from the closed-form solution of the system’s inherent generalized partial differential equation, which makes this technique physics-guided. This method applies to any type of vibration-based measurement, i.e., displacement, velocity, acceleration, and strain responses.

In this technique, the minimum sensor number, obtained from the SVD of the physics-guided basis matrix, need not be optimally placed; instead, random placement would result in sufficiently accurate full-field motion. However, in some scenarios, two or more actual sensors may be located adjacently. To avoid such a situation, the sensor location can be obtained using stratified sampling techniques such as Latin Hypercube Sampling (LHS). The full-field response obtained from a few sensors from the proposed framework can pinpoint the location of maximum vibration response; hence, this framework can be utilized to find potential damage location. This method is invariant with respect to a number of forcing functions as well as their locations.

We demonstrated the application of the proposed algorithm on the numerically simulated and experimentally validated Linear Time-Invariant (LTI) 1D and 2D structures —the Linear Time-Varying (LTV) system is for future work. For all these cases, the relative errors obtained from the proposed framework are extremely minute—such errors in full-field spatiotemporal response estimation are practically negligible; hence, this technique can be adopted in real-time structural control and health monitoring of any civil/mechanical/aerospace structure in their operating condition, leading the system to have better maintenance and resilience.

## Figures and Tables

**Figure 1 sensors-23-00384-f001:**
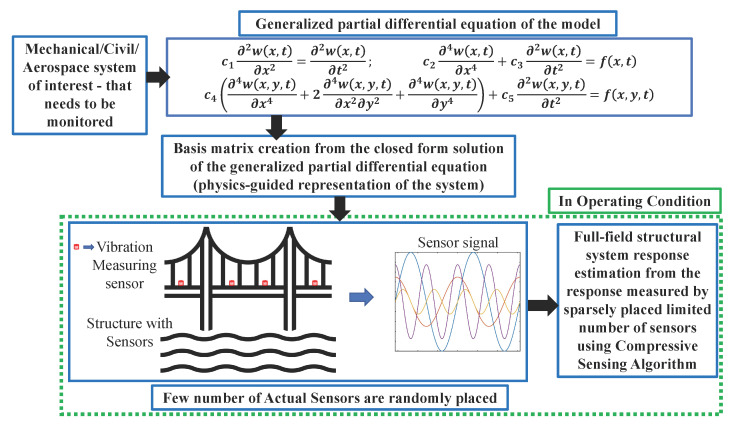
Proposed Methodology. The spatial signal basis functions are obtained from the closed form solution of the generalized partial differential equation of the model (the equations are of transverse vibration of string, beam, and plate, respectively). In operating conditions, the dense responses can be obtained from the responses of the randomly located sensors using Compressive Sensing.

**Figure 2 sensors-23-00384-f002:**
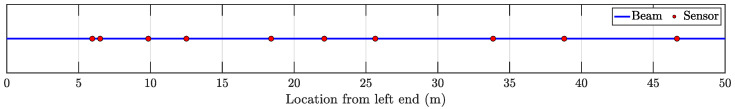
Simulation on the simply supported beam.

**Figure 3 sensors-23-00384-f003:**
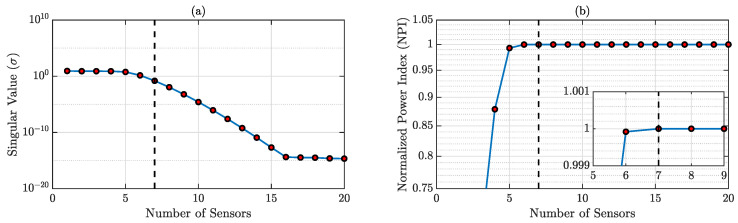
(**a**) SVD of the physics-guided basis matrix; (**b**) NPIs for different sensor numbers. NPI → 1 when the sensor number is 7 (shown in the zoomed-in version).

**Figure 4 sensors-23-00384-f004:**
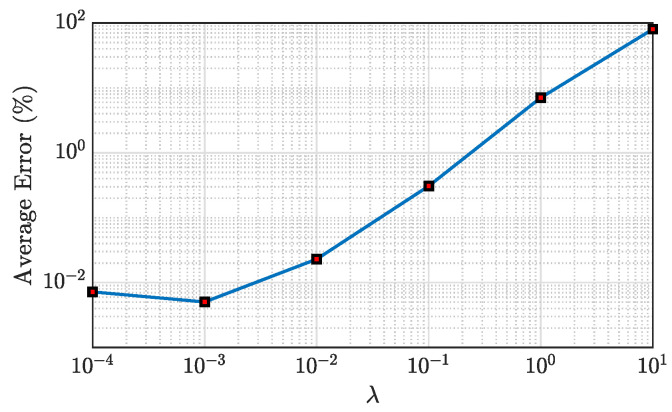
Determining the tuned λ in the CS optimization for simply supported beam.

**Figure 5 sensors-23-00384-f005:**
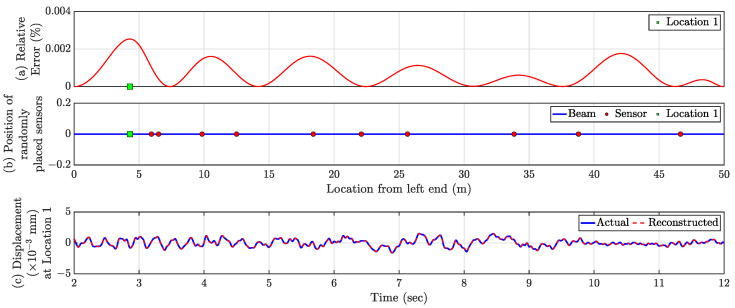
(**a**) Relative Errors ($\epsilon_{i}$) in % for the virtual dense sensing points. The ‘green square’ marker represents the location of maximum relative error. (**b**) Location of actual sensors (shown as ‘red circles’). (**c**) Comparison of actual and reconstructed time history response at Location 1.

**Figure 6 sensors-23-00384-f006:**
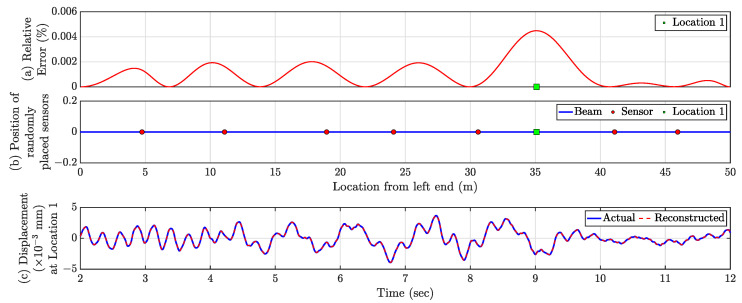
(**a**) Relative Errors ($\epsilon_{i}$) in % for the virtual dense sensing points with the optimal number of sensors placed in a stratified random fashion. The ‘green square’ marker represents the location of maximum relative error. (**b**) Location of actual sensors (shown as ‘red circles’). (**c**) Comparison of actual and reconstructed time history response at Location 1.

**Figure 7 sensors-23-00384-f007:**
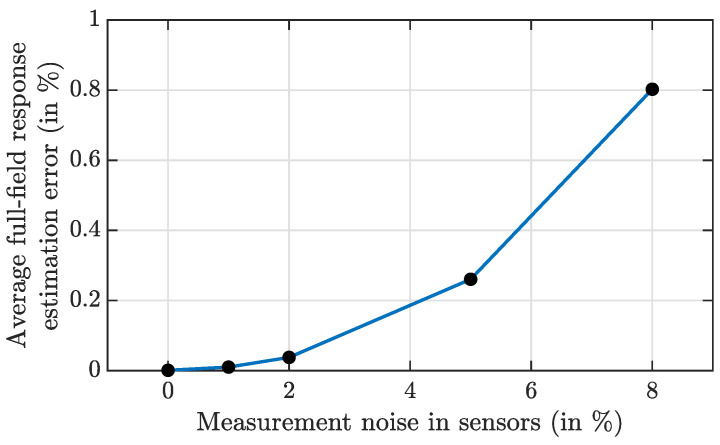
Average full-field response estimation error for the measurement noises.

**Figure 8 sensors-23-00384-f008:**
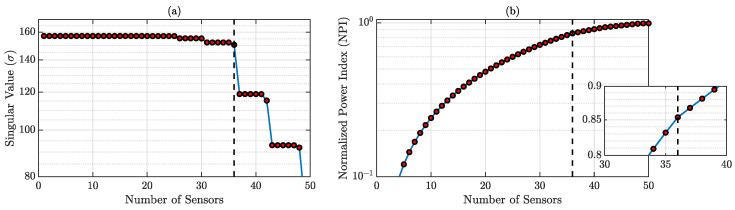
(**a**) SVs of the physics-guided basis matrix; the most dominant values are until 36; (**b**) NPIs for different sensor numbers. NPI = 0.8537 when the sensor number is 36 (shown in the zoomed-in version).

**Figure 9 sensors-23-00384-f009:**
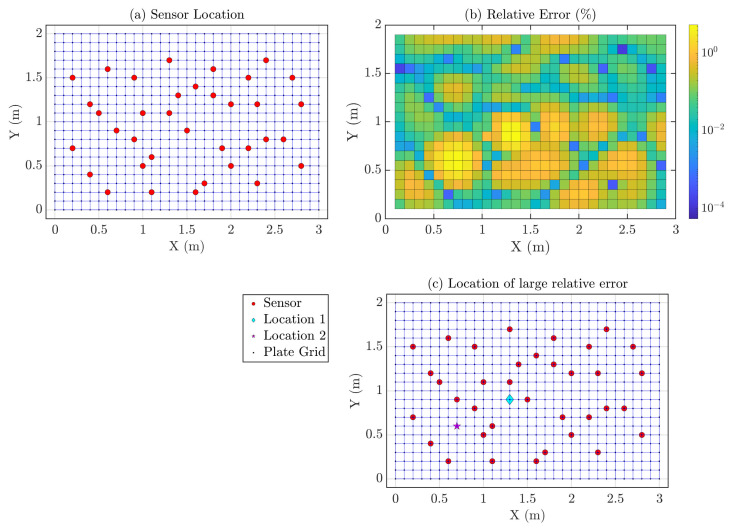
(**a**) Location of 36 stratified randomly placed sensors (shown as ‘red circles’) on the simply supported rectangular plate; the objective is to estimate the responses of all the grid points (‘blue intersections’); (**b**) Relative Errors ($\epsilon_{i}$) in % for the virtual dense sensing points; (**c**) Locations of interest with comparatively large relative errors (%). Such locations are denoted as ‘cyan diamond’ and ‘magenta star’.

**Figure 10 sensors-23-00384-f010:**
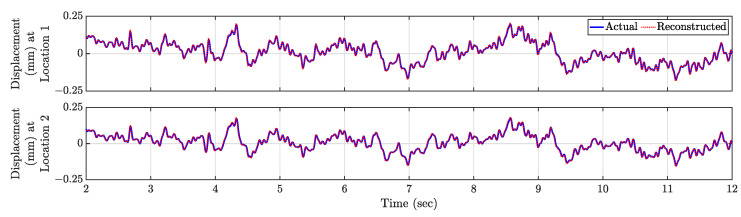
Comparison of actual and reconstructed time history response at Locations 1 (‘cyan diamond’) and 2 (‘magenta star’) in Figure 9c.

**Figure 11 sensors-23-00384-f011:**
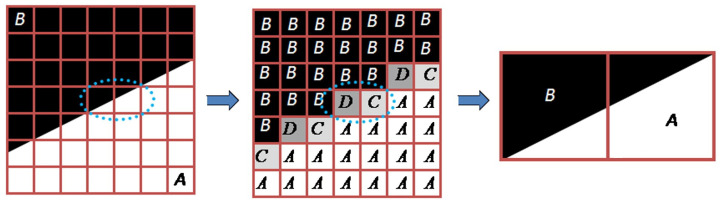
Underlying concept for estimating the edge at the subpixel level from the partial area effect [48]. Edge pixels are considered to have a weighted average intensity. Here, the pixel with value *D* has more weightage of pixel intensity *B* compared to pixel intensity *A*. Similarly, the pixels with intensity value *C* have a larger weightage of *A* compared to *B*. From these intensity values, subpixel edge locations are estimated.

**Figure 12 sensors-23-00384-f012:**
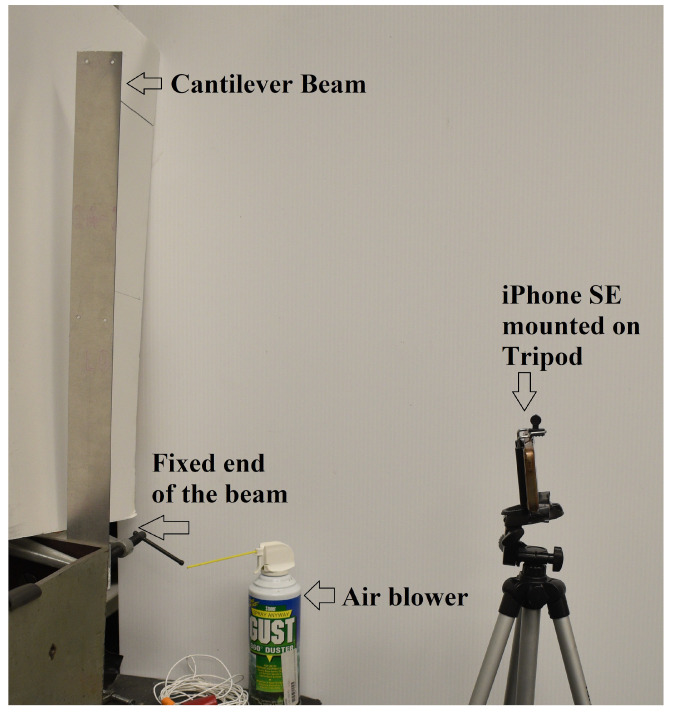
Experimental Set-up. An Aluminum beam is clamped at the table end, simulating the fixed end. This cantilever beam vibrates when an air blower blows the air on the beam. The iPhone SE records the beam vibration in slow-motion mode.

**Figure 13 sensors-23-00384-f013:**
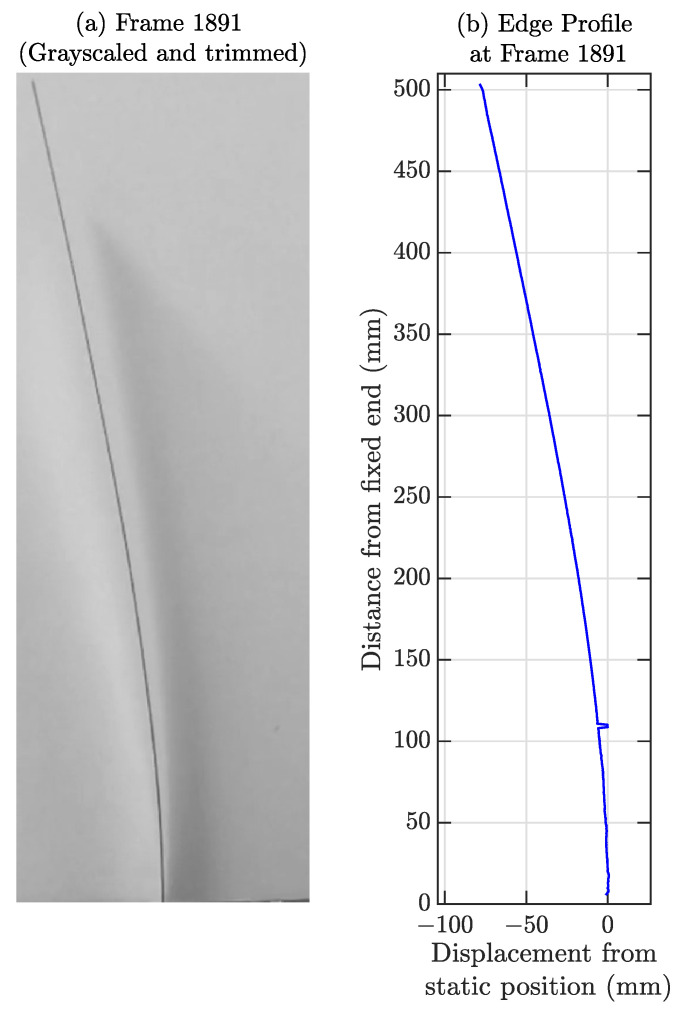
(**a**) Image frame and (**b**) corresponding edge profile for a sample frame (1 inch = 25.4 mm).

**Figure 14 sensors-23-00384-f014:**
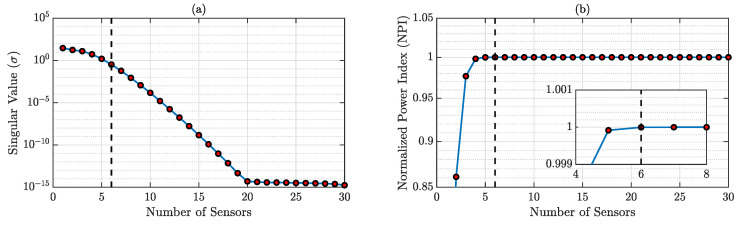
(**a**) SVs of physics-guided basis matrix, (**b**) NPIs for different sensor numbers. NPI → 1 when there are 6 sensors (shown in the zoomed-in version).

**Figure 15 sensors-23-00384-f015:**
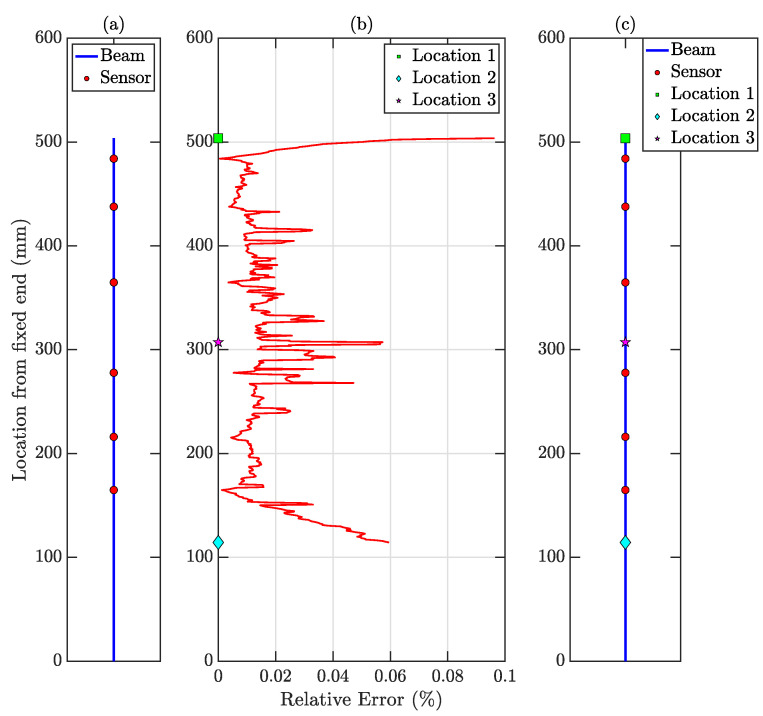
(**a**) Location of 6 Stratified randomly placed sensors (shown as ‘red circles’) in the top 15.45 inches of the cantilever beam. (**b**) Relative Errors ($\epsilon_{i}$) in % for the virtual dense sensing points (image pixels) in full-field response history estimation for the cantilever beam. (**c**) Locations of interest with comparatively large relative errors (%). Such locations are denoted as ‘cyan diamond’, ‘green square’, and ‘magenta star’.

**Figure 16 sensors-23-00384-f016:**
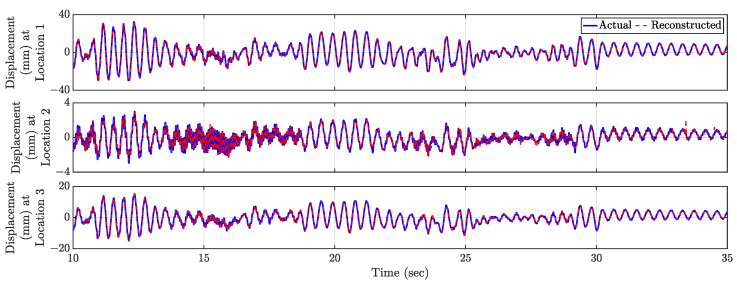
Comparison of actual and reconstructed time history response at Locations 1 (‘cyan diamond’), 2 (‘green square’), and 3 (‘magenta star’) in Figure 15.

**Figure 17 sensors-23-00384-f017:**
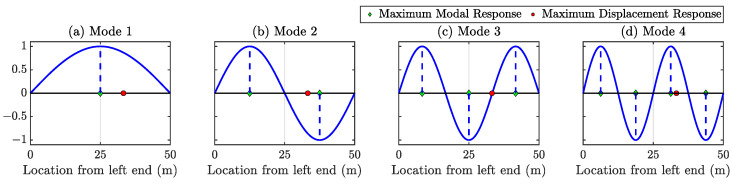
Location of the maximum displacement response is denoted as a ‘red circle’—which is impossible to obtain without the dense sensing. Even the knowledge of the maximum modal responses of different modes is not enough to find the location of the maximum displacement response. The location of maximum amplitude is very close to the node point of mode 3.

## Data Availability

Data will be made available on request.

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
