# Peer review of "Physics-Guided Real-Time Full-Field Vibration Response Estimation from Sparse Measurements Using Compressive Sensing"

_sensors, 2022, doi:10.3390/s23010384_

Round 1
Reviewer 1 Report
The authors proposed a full-field measurement method using physics-informed closed-form solution of the generalized partial differential equation of the system. It is a very interesting work. Overall, this paper is written very well. There are just some minor comments.
1). The abstract and conclusion can be shortened. Some contents of the abstract and conclusion are overlapped.
2). Following paper can be a good reference for this topic; Modal identification of simple structures with high-speed video using motion magnification.
Reviewer 2 Report
The paper
“Physics-guided real-time full-field vibration response estimation from sparse measurements using Compressive Sensing”,
by Debasish Jana & Satish Nagarajaiah,
discusses an approach for the real-time, full-field measurement of the vibration response of a target structure by means of Compressive Sensing to compensate for the use of a limited number of (physically attached or not) output channels.
This research article paper could be of great interest to researchers in the field of Structural Health Monitoring, in particular concerning real-time and/or video-based methodologies.
The Motivations are clearly stated and understandable; the procedure itself seems to be valid and has been validated both numerically and experimentally. However, the paper needs some improvements, both in its content and presentation; these are enlisted here below.
1. The abstract is quite too long and should be reduced to 200- max 250 words, for conciseness’ sake, relocating all the details to the Introduction (e.g. the discussion of DIC is unnecessary in the abstract since the technique is not actually used in the paper).
2. The Authors refer multiple times in their text to their previous work https://doi.org/10.1016/j.engstruct.2022.115280. However, the main novelties and differences between this work and that previous research should be clearly stated and emphasised in the Introduction.
3. Figure 1 should be more properly introduced in the text as the flowchart of the proposed methodology.
4. This paper focused on data sparsity in the spatial domain (which makes sense due to the practical limitation on the number of available sensors on large structures and infrastructures). However, would the proposed approach also be viable for data sparse in the time domain, e.g. due to very low sampling frequency and/or gaps in the data acquisition? This would require a different set of basis functions.
5. Related to the previous remark, to what extent can the method be generalized? As it is now, to be physics-guided, it requires a certain knowledge of the expected spatial basis functions. This is easily done for beam- and plate-like structures (especially considering the EB and Kirchhoff models). However, these functions might not be easily guessable for very complex, unconventional 3D structures.
6. The numerical tests should also investigate the robustness to measurement noise of the proposed approach. Thus, different noise levels should be tested (artificially adding white Gaussian noise in the simulated data to decrease the SNR)
7. In Eq (1), the cross-sectional area A and inertia I are reported as a function of x, i.e. in their most general formulation as a variable along the beam main axis. However, from the following discussion and calculations, it seems that the specific case of a constant cross-section has been used.
8. The Authors clearly state that “this technique can be adopted in real-time structural control and health monitoring of any civil/mechanical/aerospace structure in their operating condition”. Hence, it would be worth expanding the discussion (perhaps in the Introduction) about the potential applications of real-time Structural Health and Condition Monitoring to structures and rotating machinery, referring to the many examples available in the existing scientific literature, e.g. https://doi.org/10.3390/buildings12030310.
9. At many points throughout the text, percentage results are reported up to the fourth decimal digit (e.g. 5.1336% and 4.4852%). This is a bit too unrealistic; please consider using fewer digits.
10. At the beginning of Section 5, the Authors correctly report that video-based monitoring in general (and edge detection in particular) allow monitoring large structures/infrastructures with one device rather than a large array of attached sensors. That aspect could be further expanded, also referring to the existing scientific literature in the field of damage detection and localisation in operational deflection shapes.
11. Please double-check your text for grammar mistakes and typos. E.g. page 5 line 184, "Length" should be all lowercase letters.
12. The last section is generally named “Conclusions” rather than “Concluding Remarks”.
Round 2
Reviewer 2 Report
This Reviewer is satisfied with the detailed response to the first round of comments and with the current form of the paper. Full acceptance is suggested.